# Metabolic, Organoleptic and Transcriptomic Impact of *Saccharomyces cerevisiae* Genes Involved in the Biosynthesis of Linear and Substituted Esters

**DOI:** 10.3390/ijms22084026

**Published:** 2021-04-14

**Authors:** Philippe Marullo, Marine Trujillo, Rémy Viannais, Lucas Hercman, Sabine Guillaumie, Benoit Colonna-Ceccaldi, Warren Albertin, Jean-Christophe Barbe

**Affiliations:** 1University Bordeaux, ISVV, Unité de Recherche Œnologie EA 4577, USC 1366 INRA, Bordeaux INP, F-33140 Villenave d’Ornon, France; marine.trujillo@hotmail.fr (M.T.); remyvi17@gmail.com (R.V.); lucas.hercman@hotmail.fr (L.H.); warren.albertin@u-bordeaux.fr (W.A.); 2Biolaffort, 11 Rue Aristide Bergès, F-33270 Floirac, France; 3Pernod Ricard, 51 Chemin des Mèches, F-94000 Créteil, France; benoit.colonna@laposte.net; 4University Bordeaux, ISVV, UMR 1287 Ecophysiologie et Génomique Fonctionnelle de la Vigne, 210 Chemin de Leysotte, F-33140 Villenave d’Ornon, France; sabine.guillaumie@inrae.fr

**Keywords:** substituted ester metabolism, wine fermentation, *MGL2*, *YJU3*, histone acetylation

## Abstract

Esters constitute a broad family of volatile compounds impacting the organoleptic properties of many beverages, including wine and beer. They can be classified according to their chemical structure. Higher alcohol acetates differ from fatty acid ethyl esters, whereas a third group, substituted ethyl esters, contributes to the fruitiness of red wines. Derived from yeast metabolism, the biosynthesis of higher alcohol acetates and fatty acid ethyl esters has been widely investigated at the enzymatic and genetic levels. As previously reported, two pairs of esterases, respectively encoded by the paralogue genes *ATF1* and *ATF2*, and *EEB1* and *EHT1*, are mostly involved in the biosynthesis of higher alcohol acetates and fatty acid ethyl esters. These esterases have a moderate effect on the biosynthesis of substituted ethyl esters, which depend on mono-acyl lipases encoded by *MGL2* and *YJU3*. The functional characterization of such genes helps to improve our understanding of substituted ester metabolism in the context of wine alcohol fermentation. In order to evaluate the overall sensorial impact of esters, we attempted to produce young red wines without esters by generating a multiple esterase-free strain (Δ*atf1*, Δ*atf2*, Δ*eeb1*, and Δ*eht1*). Surprisingly, it was not possible to obtain the deletion of *MGL2* in the Δ*atf1*/Δ*atf2/*Δ*eeb1/*Δ*eht1* background, highlighting unsuspected genetic incompatibilities between *ATF1* and *MGL2*. A preliminary RNA-seq analysis depicted the overall effect of the Δ*atf1*/Δ*atf2/*Δ*eeb1/*Δ*eht1* genotype that triggers the expression shift of 1124 genes involved in nitrogen and lipid metabolism, but also chromatin organization and histone acetylation. These findings reveal unsuspected regulatory roles of ester metabolism in genome expression for the first time.

## 1. Introduction

*Saccharomyces cerevisiae* is the main yeast species involved in the alcoholic fermentation of many beverages and foods, including bread, beer, wine, and sake [1]. The secondary metabolism of fermenting yeast is the source of a broad range of volatile compounds [2,3] that contribute to the complex flavors of fermented beverages [4,5,6,7]. Volatile esters represent a noteworthy chemical family that has been widely investigated in different beverages, since they confer a wide palette of fruity notes. Since they are connected to the carbon and nitrogen metabolism of fermenting yeast, a large number of genetic variations modulates the biosynthesis of esters. In this context, many quantitative genetics programs have been carried out to decipher the phenotypic variability among strains [8,9,10]. The key enzymatic activities and the surrounding biochemical pathways involved in ester synthesis have been widely reviewed [11,12]. Higher alcohol acetates (HAA) result from the enzymatic condensation of acetyl-CoA and higher alcohols, which are derived from amino acid catabolism through the Ehrlich pathway [13,14,15]. This reaction is catalyzed by alcohol acetyl transferases (AAT = EC 2.3.1.84) [16,17] encoded by the genes *ATF1* and *ATF2* [18,19]. According to several authors, the protein *Atf1p* is the most important for the production of acetate esters [18,20,21]. More recently, a mitochondrial ethanol acetyltransferase Eat1p belonging to the AAT family was characterized for its contribution to ethyl acetate production [20]. Fatty acid ethyl esters (FAEE) result from the condensation of an acyl-CoA component with ethanol [17,21]. This ester family is synthetized by three enzymes showing a moderate sequence divergence (*Eht1p*, *Eeb1p*, and *Mgl2p*). *Eht1p* and *Eeb1p* are acyl-CoA:ethanol-O-acyl transferase (AEATase = EC 2.3.1.75) and contribute to FAEE synthesis in a synthetic medium. The enzymatic activity of *Eht1p* has been also validated in vitro by GC-MS analyses [22]. Although sharing a high sequence homology, the third protein (*Mgl2p*) has a slighter impact on ethyl ester biosynthesis [21].

Alongside these two families, substituted ethyl esters have been also identified in fermented beverages. This third group is defined by the presence of substituted chains that may be alkylated and/or hydroxylated. Alkylated ethyl esters (AEE) (ethyl 2-methylpropanoate, ethyl 2-methylbutanoate, ethyl 3-methylbutanoate) result from the esterification of ethanol and alkyl acid derived from the Ehrlich pathway [23,24]. Hydroxylated ethyl esters (HEE) such as ethyl 2-hydroxy-4-methylpentanoate and ethyl 3-hydroxybutanoate have also been described [23,25]. Compared to their linear counterparts, substituted esters are produced in smaller quantities by *Saccharomyces cerevisiae*. However, their aromatic concentration steadily increases during wine aging due to the chemical esterification by ethanol of their corresponding acids, which are also produced by yeast metabolism [23,25]. These molecules have an important sensorial impact on red wines, as they enhance fruity notes thanks to perceptive interaction phenomena [25]. The biosynthesis of substituted esters has been poorly investigated. Recently, a Quantitative Trait Loci (QTL) mapping analysis revealed that the production level of ethyl 3-methylbutanoate and ethyl 2-methylpropanoate depends on multigenetic factors that modulate the biosynthesis of their metabolic precursors [10]. However, enzymatic activity controlling the esterification step has not yet been established.

This study aims to determine which *S. cerevisae* enzymes control substituted ester biosynthesis by using a functional genetics approach narrowing the genes *ATF1*, *ATF2*, *EEB1*, *EHT1*, *MGL2*, and *YJU3*. The deletion of such genes did not impact the ongoing alcoholic fermentation but did modulate the bioproduction of the different classes of esters investigated. The impact of several combinations of gene deletions was evaluated by implementing analytical chemistry in different red grape juices. In the second phase, we sought to construct an “esterase free” yeast strain in order to evaluate the sensory consequences of ester depletion in wine. Our findings suggest that unsuspected genetic interactions impair the construction of such a strain. In order to understand this surprising result, a comparative transcriptomic analysis was carried out using an RNA-seq approach, which revealed that the combined deletion of Δ*atf1*, Δ*atf2*, Δ*eeb1*, and Δ*eht1* triggers a wide transcriptomic repatterning.

## 2. Results

### 2.1. Validation of the Role of AATase and AEATase in a Red Wine Fermentation

All volatile compounds assayed, their chemical family, and their relative abbreviations are listed in Table 1 and were measured according to methods previously developed in the laboratory [25]. As described in these communications, such molecules have an effective contribution to the sensory complexity of red wines and were quantified at the end of the alcoholic fermentation.

The impact of acyl-CoA:ethanol-O-acyl transferase (Ehb1p, Eeb1p) and alcohol acetyl transferases (Atf1p, Atf2p) in an enological context was reassessed by testing the effect of the deletion of *EEB1*, *EHT1*, *ATF1*, and *ATF2* in a Cabernet Sauvignon grape must. The role of the gene *EAT1*, which controls the production of ethyl acetate in beer fermentation [20], was not assayed, since this compound was not detected in the wines analyzed. The effect of double deletions was also estimated by obtaining the strains Fx10-ΔA12 (*Δatf1*, Δ*atf2*) and Fx10-ΔE12 (Δ*eeb1*, Δ*eht1*) (Table 2). At the end of the alcoholic fermentation, 18 volatile compounds were quantified by GC-MC and the results are detailed in Appendix A. This set of volatile compounds encompassed four higher alcohol acetates (HAA), six fatty acid ethyl esters (FAEE), four alkylated ethyl esters (AEE), and four higher alcohols (HA). Gene deletion did not impact the fermentation kinetics of the strain Fx10 and all the resulting wines reached similar values of residual sugar and acetic acid production (Appendix A). In order to allow an easier comparison of gene deletions, the data were normalized by the average value of the control strain (Fx10). Each ester family showed a similar variation pattern according to the gene deleted (Figure 1a). The relative production levels of representative compounds are given in Figure 1b,e. HAA synthesis was strongly reduced (80 to 95%) in the Fx10-ΔA12 strain. As previously reported [20], the most impactful enzyme was *Atf1p*, since the inactivation of *Atf2p* did not significantly impact HAA production, such as that of isoamyl acetate (C2iC5) (Figure 1b). Similarly, the esterification of C6-C12 fatty acids in FAEE was strongly reduced in the strain Fx10-ΔE12 which is deleted for the *EEB1* and *EHT1* genes. The main contributing enzyme was *Eeb1p*, which accounted for the majority of FAEE biosynthesis, as illustrated for C6C2 (Figure 1c). In contrast, AEATase activity made a minor contribution in the esterification of short fatty acid (C3 and C4), since the C3C2 concentration was not affected and the C4C2 production was only 35% reduced in the Fx10-ΔE12 strain.

The production of alkylated ethyl esters (AEE) was impacted differently. Strains with a reduced AATase activity (Atf1p, Atf2p) produced fewer AEE compounds, with a drop of nearly 40% for 3mC4C2 and 2mC4C2 (Figure 1d,e, respectively). The contribution of AEATase (Eeb1p, Eht1p) was more contrasted. Whereas the inactivation of Eht1p reduced the production of 3mC4C2 (Figure 1e), the inactivation of Eeb1p enhanced the production of 2mC4C2 and PhC2C2 (Figure 1a,d), suggesting a different contribution of this enzyme to the esterification of alkyl substituted acids.

### 2.2. Functional Characterization of a Nearly-Esterase-Free Strain

We then constructed a quadruple deleted strain (Δ*atf1*, Δ*atf2*, Δ*eeb1*, Δ*eht1*) by crossing haploid segregants of Fx10-ΔE12 and Fx10-ΔA12 (Table 2). The resulting strain, Fx10-ΔAE, was isogenic to Fx10 but lacked AATase and AEATase activities. Both strains were fermented in two macerated red juices (Merlot and Tempranillo) containing an 8:2 mix of grape juice and skins. This more complex matrix better mimicked the conditions of winemaking. At the end of the fermentation, 31 compounds were quantified by GC-MS and GC-FID (Appendix A), including 18 esters, six HAA, six FAEE, four AEE, and two HEE (hydroxylated ethyl esters), as well as 13 of their corresponding alcohols or acids: four HA (higher alcohols), four VAc (volatile acids), three AAc (alkylated acids), and two HAc (hydroxylated acids). All these molecules were produced by yeast metabolism, since they were not detected in the grape must (data not shown). The impact of the *strain* and *must* factors and their possible interaction was evaluated by a two-way analysis of variance (ANOVA α = 0.001) (Appendix A). Most of the phenotypic variability observed was due to the strain effect, since the grape juice origin significantly impacted only two compounds (C2iC5 and 2h4mC5). This analysis confirmed that linear ester biosynthesis (except that of ethyl propanoate) is strongly reduced in the Fx10-ΔAE strain (Figure 2a), in agreement with the results presented Figure 1. Interestingly, the combined depletion of Atf1p, Atf2p, Eeb1p, and Eht1p slightly enhanced the production of AEE (Figure 2a). In this trial, hydroxylated ethyl esters (HEE) were also assayed using the procedure described by Lytra et al. (2017). The strain Fx10-ΔAE showed a drastic reduction (−90%) in ethyl 3-hydroxy butanoate (3hC4C2) in both grape juices, whereas its production of ethyl-leucate (2h4mC5C2) was not significantly impacted (Figure 2a).

The final concentrations of many ester-metabolic precursors were impacted differently (Figure 2b). The production of higher alcohols (HA), linear (VAc), and substituted acids (AAc and HAc) were modified moderately, suggesting that gene deletion affects the esterification level of these molecules but not their biosynthesis. This could be explained by the fact that most volatile acids and higher alcohols are quantitatively much more abundant that their relative esters. In addition, most of them are derived from the same alpha keto acid and their concentrations are likely buffered by oxidoreductive reactions. In contrast, biosynthesis of hexanoic and octanoic acids was strongly reduced (−90%), supporting the idea that the drop in C6C2 and C8C2 observed was coupled with their precursor synthesis. Finally, the production of hydroxylated acids was enhanced, especially for 3-hydroxy butanoic acid, which was increased twofold in wines fermented by the Fx10-ΔAE strain (Figure 2b).

The biosynthesis pathways of hydroxylated acids and their relative esters have not been described before. Since they play a critical role in the evolution of the fruity notes of red wines, we sought to determine which esterase activity is involved in their variation. According to Saerens et al. (2006), a third protein (Mgl2p) may also have acyl-CoA:ethanol-O-acyl transferase activity in *S. cerevisiae*. By deleting the *MGL2* gene in the Fx10 strain and crossing it back with the Fx10-ΔE12, we obtained the triple deleted strain Fx10-ΔME (Δ*mgl2*, Δ*eeb1*, Δ*eht1*), supposed to be lacking in any AEATase activity (Table 2). The production level of HEE was compared to the strain Fx10-ΔAE and the *wt* strain Fx10 (Appendix A). A drop in 3hC4C2 was observed in both mutants, demonstrating that this ester is mostly produced by AEATase. In contrast, the production of ethyl-leucate (2h4mC5C2) was only slightly reduced (−15%) in respect to the wild type, suggesting that other enzymatic activities are involved in the biosynthesis of this ester, which is hydroxylated and alkylated.

### 2.3. Sensory Profiling of a Wine with Reduced Ester Content

The Merlot and Tempranillo wines produced in this experiment provided the opportunity to analyze the organoleptic impact of ester depletion in controlled conditions for the first time. Different descriptors related to red wine fruity perception were explored through sensorial analysis (see methods). The panelist found a significant intensity variation for the overall aroma (OA), fermentative aroma (FA), red fruits (RF), and fresh fruits (FF) descriptors. In contrast, the black fruits (BF) and jammy fruits (JF) notes were not significantly impacted (Wilcoxon test, α = 0.05) (Figure 3). This result demonstrates that the cumulative depletion of *Atf1p*, *Atf2p*, *Eeb1p*, and *Eht1p* strongly decreases the fruity aromatic notes of red wine in connection with the concentration drop in linear and some substituted esters (Figure 2a). Since the production of other molecules (not assayed) might also have an effect on fruity aroma, we completed this sensorial analysis by an aromatic reconstitution. Wines fermented with Fx10-ΔAE and Fx10 strains were supplemented in various esters up to the same level of ester in the supplemented wines (Appendix A). A second panelist used triangular tests to confirm that the supplemented wines did not show significative differences with the control. This result confirms that the ester production discrepancy was the unique cause of the sensorial differences observed between the tasted wines.

### 2.4. Functional Characterization of Mgl2p and Yju3p, Two Mono-Acyl Glycerol Lipases Involved in the Synthesis of Substituted Esters

In the previous sections we clarified the role of AATses and AEATases in the biosynthesis of linear esters and 3hC4C2. However, AEE and ethyl leucate biosynthesis have not yet been completely elucidated. As well as *Atf1p*, *Atf2p*, *Eeb1p*, and *Eht1p*, other enzymes may play a role in substituted ester biosynthesis. We hypothesized that the two proteins *Mgl2p* and *Yju3p* may modulate the concentration of fatty acids, which are the precursors of ethyl esters. Indeed, these proteins have been characterized for their mono-acyl glycerol lipase (MAGLase) activity [26,27]. To test this hypothesis, the same functional genetic strategy was applied and both enzymes were inactivated in the strains Fx10-ΔM (Δ*mgl2*) and Fx10-ΔY (Δ*yju3*). Their combined effect was also evaluated in the double mutant Fx10-ΔYM (Δ*yju3*, Δ*mgl2*) (Table 2). Again, the fermentation kinetics of such strains was similar to the control (data not shown). The depletion effect of MAGLase activity is summarized in a heat map (Figure 4a). Mgl2p and Yju3p played a very moderate role in the biosynthesis of linear esters of fatty acids and did not influence the production level of higher alcohols and their corresponding acetate esters (Appendix A). Although significant, the inactivation of MAGLases had only a slight effect on FAEE production (−10%) (Appendix A) compared to AEATse inactivation (Figure 1 and Figure 2). In contrast, MAGLases were significantly involved in the de novo synthesis of most substituted ethyl esters. Indeed, AEE production was reduced by nearly 50% compared to the control in single and double mutants (Kruskal–Wallis test α = 0.01) (Figure 4b). However, the double deletion of *MGL2* and *YJU3* did not abolish AEE production, suggesting that other enzymes (including *Eeb1p* and *Eth1p*) compensated for their inactivation. Remarkably, *Mgl2p* and *Yju3p* did not affect the production level of a phenyl-substituted ester (PhC2C2), which seemed to have been produced by another enzyme (Figure 4a). The biosynthesis of hydroxylated esters showed a more contrasted genetic determinism. Ethyl-leucate (2h4mC5C2) was reduced by nearly 50% in the double mutant (Figure 4c) but, as observed for alkylated esters, its production was not fully abolished. This drop was positively coupled with the biosynthesis of its corresponding acid (2h4mC5) (Figure 4c). This suggests that MAGL may control the biosynthesis of the ethyl 2-hydroxy-4-methylpentanoic acid, which in turn influences the production level of ethyl-leucate. In contrast, the production level of ethyl-3 hydroxy-butanoate (3hC4C2) was not evenly impacted by MAGL inactivation. *MGL2* deletion promoted 3hC4C2 synthesis (+50%), whereas *YJU3* deletion had no impact with a dominant effect on *MGL2* (Figure 4c). Altogether, these findings demonstrate for the first time the impact of MAGLase activity on the biosynthesis of volatile substituted esters in the context of alcoholic fermentation.

### 2.5. Attempts to Construct a Fully Esterase-Free Yeast Strain

The possible metabolic connection between enzymatic activities (AATase, AEATase, and MAGLase) prompted us to integrate the *MGL2::KanMx* allele into the Fx10-ΔAE strain, which significantly impacts AEE and HEE biosynthesis. Using a breeding strategy, an F1 hybrid (Fx10-ΔAEM) was obtained by crossing the appropriate haploid segregants of Fx10-ΔAE and Fx10-ΔM. Since deleted genes are located in different chromosomes, the expected frequency of an Fx10-ΔAEM progeny with the quintuple-deleted genes would be 1/32. Surprisingly, by genotyping such progeny by PCR, we failed to identify any segregant harboring all five deleted genes in the 296 spores dissected (Figure 5a). In order to increase the chance of obtaining this progeny, different combinations of Fx10-ΔAEM segregants were crossed with each with the aim of fixing some desired alleles. For some haplotype combinations, the isolated F2-zygotes did not develop a central bud and stopped their growth at the first division stage. This prezygotic incompatibility was visually observed in three haplotype combinations (H1xH2, H1xH3, and H3xH6) in 70 distinct zygotes (Table 3). In contrast, other crosses (H1xH5 and H1xH6) developed zygotes with a perfect fitness. The sporulation of these viable F2-hybrids (H1xH5 and H1xH6) would also allow the isolation of segregants carrying the five deleted genes with a frequency of 1/8. Among the 176 progenies dissected, we failed to obtain a strain deleted for the five genes (Δ*atf1*, Δ*atf2*, Δ*eeb1*, Δ*eht1*, Δ*mgl2*). In these two F2-hybrids, seven pairs of double mutants were expected with a frequency of 50% (Figure 5b). Surprisingly, the double mutant (Δ*atf1*, Δ*mgl2*) was barely represented (5%) compared to other combination pairs (chi-square test *p* = 1·10^−7^), indicating a probable deleterious interaction. However, this interaction cannot be considered a synthetic lethality since some (Δ*atf1*, Δ*mgl2*) strains were isolated in the Fx10-ΔAEM progeny (haplotypes H2 and H3). Another noteworthy result is the low germination percentage of the hybrids dissected (between 82 and 75%), which is quite unusual in nearly isogenic crosses. By typing all the progenies dissected, we inferred the genotype of non-viable clones. Most of them appeared to have the “five deleted genes” pattern sought (data not shown). These findings strongly suggest that the combined loss of *MGL2* and *ATF1* genes confers a drastic and unexplained loss of viability.

### 2.6. Transcriptomic Analysis of the Fx10-ΔAE Strain Reveals Unsuspected Consequences of the Depletion of Esterase Activities

The genetic incompatibility revealed in the previous section contrasted with the absence of macroscopic phenotypes in strains deleted for one or many esterase genes. For instance, the strain Fx10-ΔAE (Δ*atf1*, Δ*atf2*, Δ*eeb1*, Δ*eht1*) showed very similar fermentation kinetics to Fx10 (data not shown). This surprising result prompted us to investigate in depth the physiological consequences of multiple gene deletion via a transcriptomic approach. Biomass of strains Fx10-ΔAE and Fx10 were collected at 30% of the alcoholic fermentation (Merlot grape juice), total RNA was extracted, and the corresponding cDNAs were sequenced using proton technology (see Methods). RNA sequencing allowed the quantification of 6287 commonly detected genes. The average fold change ratio log2 (Fx10-ΔAE/Fx10) was computed and genes showing a statistical difference were defined by an ANOVA or Kruskal–Wallis test (whenever ANOVA assumptions were not met). This analysis identified 1124 Differentially Expressed Genes (DEG) showing a significative difference with at least a twofold change expression. This result suggests that the combined deletion of four genes triggers an extensive transcriptome remodeling. Surprisingly, a strong bias toward gene overexpression in the strain Fx10-ΔAE was found, since 1102 and 22 genes were up- and downregulated, respectively (Figure 6a and Appendix A). As expected, *ATF1*, *EHT1*, and *EEB1* were part of the shut-down genes, since very few reads were mapped at their positions. Initially, *ATF2* did not appear as a DEG (Figure 6b). On closer scrutiny it was revealed that *ATF2* overlapped partially with another ORF, *YGR176W*. In order to correctly quantify *ATF2* only, overlapping reads between *ATF2* and *YGR176W* were subtracted, revealing an actual absence of *ATF2* mRNA in the Fx10-ΔAE mutant (Figure 6b). This first analysis confirmed the correct and complete deletion of the four genes in the strain Fx10-ΔAE. Interestingly, the gene *MGL2* was identified as up-regulated in the Fx10-ΔAE mutant, suggesting the existence of dosage compensation mechanisms between esterase genes. In addition, *YJU3*, the second MAGL gene investigated in this study, was also significantly upregulated in the mutant (*p*-value < 0.05). Since it showed a small increase (44%, less than a twofold change), this gene was not considered as a DEG in further analyses.

A gene ontology (GO) analysis identified 23 GO categories that were significantly enriched or depleted in DEGs (hypergeometric distribution with Holm–Bonferroni correction, α = 0.05). Interestingly, most of the GO categories concerned major biological processes involved in cellular housekeeping functions, including mRNA and rRNA processing (GO:0006397, GO:0006364), cell division (GO:0051301), transmembrane transport (GO:0055085), and oxidoreduction process (GO:0055114). The strong enrichment in genes related to RNA metabolism was also confirmed by a significative enrichment in RNA binding function (GO: 0003723), as well by the fact that half (517) of the 1124 genes were associated with the nucleus (GO:0005634), i.e., three times more than the proportion expected for the whole genome (Appendix A). This first analysis revealed that the deletion of the main esterase genes triggered an unexpected modification of genome expression in the strain Fx10-ΔAE. In order to unravel this surprising expression repatterning, we sought functional connections between DEGs by searching in the protein–protein interactions (PPI) database (string-db.org). A cluster analysis of the PPI score matrix allowed the identification of 13 PPI clusters sharing functional relationships (Figure 7). Each cluster encompassed more than 14 proteins and reached an average score higher than 0.51. Most of them (clusters 1, 2, 6, 7, 8, 10, 11, and 12) were enriched in protein networks related to nitrogen and lipid metabolism (GO:0006807, GO:1901564, GO:1901566, GO:0006629). This connection with metabolism was not directly identified by the GO enrichment analysis, highlighting the interest of using a PPI score to discover functional components in genome-wide datasets. Interestingly, cluster 13 encompassed 14 proteins mostly involved in lipid metabolism and acyl transferase activity (GO:0016746), including the MAGLase *Mgl2p*. This result demonstrates that the lack of AATses and AEATase activities triggers the expression of many genes involved in the amino acid and lipid biosynthesis pathway.

Other PPI clusters were related to RNA regulation (clusters 1, 2, and 6, GO:0016070), mitochondrion and mitochondrial translation (clusters 5 and 10, GO:0005739, GO:0032543), cell cycle (cluster 3, GO:0007049), and the site of polarized growth (cluster 9, GO:0030427), congruently with the previous GO analysis. A few other elements were identified by PPI analysis: response to stimulus (cluster 4, GO:0050896), cellular aromatic compound metabolic process (GO:0006725), and the involvement of phosphoprotein (clusters 3, 4, and 9, KW-0597). Finally, cluster 12, particularly enriched in proteins, was involved in covalent chromatin modification (CL:3190) encompassing proteins belonging to histone acetyltransferase (CL:3659) and histone deacetylase (CL:3384) complexes (https://version-11-0b.string-db.org/cgi/network?networkId=b8KiKTUhCG3J). 

The presence of proteins related to chromatin modification and the wide overexpression pattern of the strain Fx10-ΔAE led us to check possible connections with histone modifications by searching in a dataset of nucleosome co-immuno-precipitation [28]. The average ratios of acetylation and methylation in up- and downregulated genes were compared to those of non-significant genes (Figure 8). Most histone modifications were significantly affected in the mutant, with seven out of eight modifications enriched in the pool of upregulated genes. Accordingly, downregulated genes were depleted in histone modifications (5/8). This general trend is consistent with the fact that histone modifications such as acetylation or methylation are frequently associated with increased transcriptional activities and chromatin structure. Altogether, these results underline a significant disturbance of the histone modification and chromatin structure in the Fx10-ΔAE mutant, congruent with the high number of upregulated genes in this strain.

## 3. Discussion

### 3.1. General Reassessment of the Esterase Contribution in Red Winemaking

Although esters are very impactful compounds of wine flavor, the functional characterization of genes and enzymes involved in their biosynthesis has rarely been achieved in natural grape juices. Indeed, the genetic basis of ester biosynthesis has been evaluated in culture broths [18,21] and in synthetic grape juice [19]. Since the nitrogen composition has a strong impact on ester metabolism [29,30], the use of artificial media may create physiological biases with respect to natural grape juices. This discrepancy has recently been confirmed in a comparison of the gene deletion of several genes in laboratory culture broth and in white grape juice [20]. By analyzing the deletion effect of six genes in an enological context, we clarified their role in the biosynthesis of linear and substituted esters, including HEE. In order to draw robust conclusions, gene deletion effects were evaluated in three different matrices (Cabernet Sauvignon, Merlot, and Tempranillo). The use of a fully homozygous diploid background (Fx10) allowed for the convenient construction of simple and multiple deleted strains by combining genetic engineering and classical breeding approaches. Figure 9 summarizes their relative contributions to the biosynthesis of ester classes considered in this study.

The inactivation of AATses (Atf1p and Atf2p) and AEATses (Eeb1p, Eht1p) confirmed their respective roles in the biogenesis of HAA and FAEE compounds. Both pathways were quite independent, since the depletion of one class of enzyme had a null or very moderate effect on the biosynthesis of the other group, except for the acetates of phenyl-ethanol and ethyl-butanoate, which were slightly enhanced by the inactivation of AEATses and AATses, respectively. When both enzymatic activities were inactivated (strain Fx10-ΔAE), linear ester biosynthesis was strongly reduced, especially for HAA. In contrast, the biosynthesis of ethyl propanoate (C3C2) was not affected at all. In a recent study, the production of this ester was linked to a mitochondrial ethanol acetyltransferase Eat1p. *EAT1* overexpression increases the biosynthesis of ethyl propanoate by condensing ethanol and propionyl-CoA [20]. However, according to the same authors, the deletion of this gene does not impact its biosynthesis in grape juice, likely due to a low oxygenation level.

Besides the well-documented biosynthesis of linear esters, this study focused on substituted ester metabolism, which contributes to the fruity notes of young red wines [25,31]. Our investigations demonstrated that yeast esterases (Atf1p, Atf2p, Eeb1p, Eht1p) are not the main contributors to AEE and HEE biosynthesis. Indeed, the esterification of alkyl substituted acids in enological conditions (2-methyl-propanoate, 2-methyl butanoate, and 3-methyl butanoate) was poorly impacted by AATases inactivation (−25%). Since AATases catalyze the condensation of acetyl-CoA and higher alcohols, their impact on substituted ethyl ester is likely indirect and should be due to metabolic connections existing between methylated alcohols and methylated acids that are derived from the respective reduction and oxidation of methyl-substituted aldehydes [32]. AEATases have a more contrasted role, since the production of some alkyl esters was reduced (3mC4C2) or enhanced (PhC2C2). It is important to note that the concentrations measured for these latter compounds were quite low and would not impact the sensory profile of wines. Although Eeb1p and Eht1p did not efficiently catalyze the esterification of alkylated acids, they were clearly involved in the biosynthesis of ethyl 3-hydroxy-butanoate (Figure 2 and Appendix A), which conferred red-berry and fresh-fruit notes, even at subthreshold concentrations [33].

The reassessment of substituted ethyl ester metabolism was completed by an evaluation of the role of two MAGLases. The deletion of *MGL2* and *YJU3* reduced the biosynthesis of methyl substituted esters such as 2mC3C2, 2mC4C2, 3mC4C2, and 2h4mC5C2. In the particular case of ethyl leucate biosynthesis, MAGLase inactivation also reduced the concentration of its direct precursor, 2-hydroxy-4-methyl pentanoic acid, which was not observed for the other methylated acids. This result suggests that the direct precursor of ethyl-leucate may be derived from the degradation of lipid metabolism, which is consistent with the primary function of mono acyl glycerol lipases.

To date, the sensorial consequences of ester depletion have never been investigated in red wines. By fermenting two macerated grape juices with the strain Fx10-ΔAE, we demonstrated through sensory analyses that ester depletion has a significant effect on various aromatic descriptors, as it reduces the perception of red and fresh fruit notes as well as the overall aroma of the wines tasted. An aromatic reconstruction carried out with synthetic molecules demonstrated that the drop in fruity perception observed was due to the depletion of esters itself, thus ruling out the hypothetical impact of hidden compounds not assayed.

### 3.2. Genetic and Transcriptomic Experiments Revealed Unsuspected Physiological Consequences of Esterase Activity

Unexpectedly, we failed to obtain a strain deleted for all six genes investigated. Like some colleagues [20], the construction of a quadruple deleted strain, Fx10-ΔAE (Δ*atf1*, Δ*atf2*, Δ*eeb1*, Δ*eht1*), was readily achieved with a breeding approach. However, we repeatedly failed to isolate a strain carrying a fifth deletion (Δ*mgl2*) using independent strategies (cross or segregation). Strikingly, for several haplotype combinations, none of the zygotes isolated developed a central bud or stopped their growth at the first division step, suggesting a possible deficiency in cell polarization or in bud formation. This result led us to further investigate the physiological changes provoked by the deletion of the four main esterases. At the macroscopic level, no significative change with respect to the control was observed: Both strains had a similar fitness in broth media and grape juice. In contrast, strong variations were measured at the transcriptomic levels, since more than 1100 DEGs were clearly identified. The significant overexpression of *MGL2*, and to a lesser extent of *YJU3*, suggests the existence of genetic compensation mechanisms in the context of esterase depletion. As a consequence, the deletion of *MGL2* could impair such compensations, with a synthetic lethal effect in the Fx10-ΔAE background. This deleterious effect was observed when all five genes were deleted, supporting the idea that Mgl2p, Eeb1p, Eht1p, Atf2p, and Atf1p share complementary functions. Although we did not identify a precise genetic interaction, the strong reduction in germination rate for progeny carrying the Δ*mgl2* and Δ*atf1* alleles supports the idea that these two genes may play a crucial role in yeast viability through unknown mechanisms.

In phase with this genetic incompatibility, a preliminary transcriptomic analysis confirmed that the drastic reduction in ester production was not neutral in terms of molecular physiology. Indeed, the strain Fx10-AE showed a striking expression repatterning with a strong bias toward gene overexpression (1002 vs. 22 genes). GO enrichment and PPI STRING analyses depicted a large set of activated pathways. Firstly, many genes involved in nitrogen and lipid metabolism were identified. In particular, esterase inactivation enhanced the expression of genes related to basic and aliphatic amino acid biosynthesis such as lysine (*LYS14*, *LYS20*, *LYS2*, *LYS9*), arginine (*ARG4* and *ARG8*), methionine (*MET31*), leucine (*LEU5*), and threonine (*THR4*) (Appendix A). The transport of these amino acids was also enhanced with the overexpression of basic and substituted amino acid transporters (*VBA3*, *VBA5*, *BAP3*). In addition, PPI analysis identified a small cluster of 14 genes related to lipid metabolism, including *MGL2* and *YJU3.* Most of them were involved in triglyceride homeostasis and lipid droplet formation (*CST6*, *LOA1*, TGL2, *TGL3*, *TGL5*) and were annotated for their acyl-transferase activity, which may have compensated for the deletion of *EEB1* and *EHT1*. Besides these broad metabolic adjustments, many overexpressed genes were related to central cellular functions such as cell cycle and polarization, as well as RNA binding. The detection of a small cluster of 32 proteins (cluster 12) involved in chromatin modification retained our attention. This group encompassed key components (*HAD2*, *CTI6*, *AHC2*, *EAF7*) of two protein complexes involved in histone acetylation and deacetylation, HAD1 and ADA, respectively [34]. Since all genes were overexpressed in the mutant, we hypothesized that Fx10 ΔAE had an abnormal chromatin acetylation homeostasis, explaining the expression repatterning observed. This hypothesis was corroborated by the fact that the DEG set was more subject to histone acetylation than the rest of the genome. Since histone methylation and acetylation are partially correlated, the DEG set was also more subject to histone methylation. A possible link between histone modification and ester formation is the acetyl-CoA molecule itself, which constitutes a common substrate for acetyl histone transferases and AATses. In yeast, the level of acetyl-CoA is tightly regulated by acetyl-CoA synthetase (Acs1p, Acs2p) and acetyl-CoA hydrolase (Ach1p). We hypothesized that AATses could also reduce the acetyl-CoA pool by producing acetate esters of higher alcohols. Therefore, in the Fx10-ΔAE strain, AATse inactivation would increase the nucleocytosolic pool of acetyl-CoA, triggering histone acetylation, which is consistent with an overall elevation of gene expression.

## 4. Materials and Methods

### 4.1. Culture Conditions and Classical Genetics Manipulations

All the chemical product used in this study were purchased from Sigma-Aldrich company (Lyon, France). The yeast strains in this study belonged to the *Saccharomyces cerevisiae* species and were propagated at 28 °C on YPD medium (1% yeast extract, 2% peptone, dextrose 2%) complemented with 2% agar to prepare the solid medium. *KanMx* and *HygMx* markers were selected using G418 (100 µg/mL) and hygromycin B (300 µg/mL), respectively. Sporulation was induced in ACK medium (potassium acetate 1%, agar 2%) at 24 °C for three days and free spores were obtained by a cytohelicase treatment (2 mg/L, 90 min at 30 °C). Cell mating was performed by incubating 10^5^/mL of spores and/or haploid cells in YPD for 6 h at 30 °C. Newly formed zygotes were then isolated from the mix by micromanipulation using a Singer apparatus MS200. The Mendelian segregation of deleted genes was controlled by analyzing at least four complete tetrads isolated by micromanipulation.

### 4.2. Construction of Esterase-Free Yeast Strains

A collection of nine knocked-out strains isogenic to the Fx10 background was obtained (Table 2). This strain was a commercial starter Zymaflore Fx10 (Laffort, France), which is widely used for red juice fermentation. This diploid, homothallic, and fully homozygous strain was previously used by our laboratory [35]. The deletion of the six genes in this study (*ATF1*, *ATF2*, *EEB1*, *EHT1*, *MGL2*, *YJU3*) was achieved using PCR-deletion cassettes obtained by amplifying the genomic DNA of the Euroscarf collection strains Y31674, Y34807, Y33317, Y32157, Y30796, and Y34943, respectively (Oberursel, Germany). The strain Fx10 was transformed using an optimized lithium acetate protocol [36]. All constructions were verified by an appropriate insertion PCR. In brief, the verification consisted of positively amplifying by PCR a fragment containing ~600 pb of the 5′-flanking region and the 5′ part of the *KanMx4* cassette. All the primers used for this test are listed in Appendix A. Multiple deletion strains were obtained by meiotic segregation and iterative crosses with appropriate deletion mutants. Meiotic segregants were selected based on G418 resistance and verified for both marker insertion and gene deletion using appropriate PCRs. To simplify the cross procedure, a haploid isogenic clone of Fx10 (YMP34) was used. This strain was deleted for the *HO* gene by the *ho::HygKx4* cassette, allowing easy selection of F1 hybrids with G418 resistant strains [35].

### 4.3. Fermentation Conditions

Two distinct fermentation batches were made. Simple and double deletion mutants of genes involved in linear ester biosynthesis were tested in a thermo-treated grape juice of Cabernet Sauvignon (CS) harvested in 2013 in the Bordeaux area and conserved at −20 °C. The assimilable nitrogen concentration of this grape juice was 200 mg N/L with mixed sources of 18 α-amino acids and ammonium nitrogen (60/40 ratio). The source of α-amino acids was previously described by [37]; the source of mineral nitrogen was a solution of (NH_4_)_2_SO_4_. Fermentation took place in cylindric glass vessels of 300 mL with permanent stirring (200 rpm) according to the conditions described by [38].

A second fermentation batch was made in larger volumes in order to test the organoleptic impact of gene deletion. Two different grape varieties harvested in 2015 were used: a Merlot (Bordeaux area, France) and a Tempranillo (Rioja area, Spain). In order to mimic red-grape vinification, a specific protocol was developed. Full grapes were destemmed, crushed, pressed, and both the juice and the skins were conserved separately. Before freezing, potassium metabisulfite was added to the must to reach 50 mg/L of total SO_2_. For both grape varieties, the sugar concentration was set at 230 g/L of reducing sugar by adding an equimolar amount of D-glucose and D-fructose. The assimilable nitrogen concentration was adjusted to 210 mg N/L, keeping a 66:34 balance ratio between amino acids and ammonium nitrogen source using the solution described above. Fermentations took place in 2.5 L cylindric glass flasks filled with 1.6 L of juice and with 400 mL of grape skin in order to keep a juice/solid ratio of 80:20, as is usual in oenology, and each flask was mixed twice a day to ensure a homogenous fermentation.

In all fermentation batches, the grape juice was inoculated with 1.10^6^ viable cell/mL obtained from 24 h precultures carried out in half-diluted grape must sterilized by membrane filtration (cellulose acetate 0.45 μm Sartorius Stedim Biotech, Aubagne, France). Yeast viability and concentration were estimated by flux cytometry “Cell Lab Quanta SC” (Beckman Coulter, USA, California) according to the procedure previously described [39]. Fermentation kinetics were monitored by CO_2_ release [40]. At the end of the alcoholic fermentation, the wines were collected in glass bottles and stored at 10 °C for 1 week after the addition of SO_2_ (50 mg/L).

### 4.4. Total RNA Isolation and mRNA Sequencing

The fermenting yeasts were aseptically sampled at 30% of the alcoholic fermentation to achieve RNA seq analysis. Cells were pelleted by centrifugation, washed 4 times, and frozen at −80 °C. Since the grape must was rich in polyphenols and anthocyanins, the mRNAs were extracted according to Reid’s procedure (2006) adapted for yeast. The extraction buffer contained 300 mM Tris HCl (pH 8.0), 25 mM EDTA, 2 M NaCl, 2% CTAB, 2% PVPP, 0.05% spermidine trihydrochloride, and, just prior to use, 2% β-mercapto-ethanol. The yeasts were ground to a fine powder in liquid nitrogen using a mortar and pestle. The powder was added to pre-warmed (65 °C) extraction buffer at 20 mL/g of yeast and shaken vigorously. The tubes were subsequently incubated in a 65 °C water bath for 10 min and shaken every couple of min. Mixtures were extracted twice with equal volumes chloroform:isoamyl alcohol (24:1) then centrifuged at 3500× *g* for 15 min at 4 °C. The aqueous layer was transferred to a new tube and centrifuged at 30,000× *g* for 20 min at 4 °C to remove any remaining insoluble material. To the supernatant, 0.1 vol 3 M sodium acetate (pH 5.2) and 0.6 vol isopropanol were added, mixed, and then stored at −80 °C for 30 min. Nucleic acid pellets (including any remaining carbohydrates) were collected by centrifugation at 3500× *g* for 30 min at 4 °C. The pellet was dissolved in 1 mL TE (pH 7.5) and transferred to a microcentrifuge tube. To selectively precipitate the RNA, 0.3 vol of 8 M LiCl was added and the sample was stored overnight at 4 °C. The RNA was pelleted by centrifugation at 20,000× *g* for 30 min at 4 °C then washed with ice-cold 70% EtOH, air dried, and dissolved in 50–150 μL DEPC-treated water.

The RNA concentration and 260/280 nm ratios were determined with a NanoDrop ND-1000 spectrophotometer (NanoDrop Technologies, Wilmingon, DE, USA), and 1% agarose gels were run to visualize the integrity of the RNA. To improve our ability to visually assess RNA quality, the same RNA samples were run on a TapeStation 4200 system (Agilent Technologies, Santa Clara, CA, USA), using High Sensitivity RNA ScreenTape assays. mRNA isolation was performed using a NEBNext Poly(A) mRNA Magnetic Isolation Module (New England Biolabs, Ipswich, MA, USA) and library preparation was achieved using Ion Total RNA-Seq Kit v2 (Thermo Fisher Scientific, Waltham, MA, USA), following the manufacturer’s instructions. Six barcoded samples were simultaneously sequenced on an Ion Proton System (Thermo Fisher Scientific, Waltham, MA, USA) at the Genome Transcriptome Facility of Bordeaux. Reads were deposited in the NCBI SRA database www.ncbi.nlm.nih.gov in the bioproject: PRJNA704978.

### 4.5. RNA-Seq Alignment and Quantification

For read alignment, we used a reference genome composed of the sequence of the *S. cerevisiae* laboratory strain S288c (assembly: GCF_000146045.2, www.ebi.ac.uk) manually added with the 34 genes specific to the wine strain EC1118 (assembly: GCA_000218975.1, www.ebi.ac.uk [41]. Alignment was performed using the Torrent Mapping Alignment Program (TMAP) map4 module implemented in the Ion Torrent Suite 5.0.5. Default parameters were used to generate BAM files for each sample. 

A Galaxy server [42] was used to create tool suites called a “workflow” in a simple way: Data alignment was checked and filtered using SAMtools [43,44]. Unmapped reads were eliminated, as well as reads with mapping quality <20. The number of reads per gene was counted using HTseq-count [45] with the following options: format (-f): BAM, type (-t): mRNA, ID (-i): transcript_id, quality min. (-a): 20. In this case, we used a GFF3 reference file containing features from S288c (EnsemblFungi website, *S. cerevisiae* R64-1-1 release 40) added with the 34 EC1118-specific genes. Finally, gene expression was normalized to reads per kilobase million [46] using the R software [47]. A total of 6287 genes were quantified.

### 4.6. RNA-Seq Statistical Analyses

Differentially Expressed Genes (DEG) between wildtype and mutant strains were determined using ANOVA. A Levene test was carried out beforehand to verify homoscedasticity. For the genes showing heteroscedasticity, non-parametric tests were performed (Kruskal–Wallis, *agricolae* package, R). The *p*-values were corrected for multiple tests with the method of Benjamini–Hochberg (FDR, false discovery rate). We considered differentially expressed genes with (i) *p*-values < 0.05 and (ii) higher than twofold changes (1124 genes, 22 downregulated, and 1102 upregulated in the mutant).

For gene ontology (GO) analysis, the gene_association.sgd.gz file (December 2003, http://www.geneontology.org/) was used to identify the GO terms associated with all quantified genes. Hypergeometric tests were performed to identify over- or underrepresented GO terms (functions, processes, or components) for upregulated and/or downregulated genes in the mutant. *p*-Values were corrected for multiple tests with the method of Benjamini–Hochberg with R.

Protein–protein interactions were analyzed using the STRING database v11 (http://string-db.org/) and the STRINGdb package ^®^ [48]. For each pair of DEG genes, a protein–protein interaction (PPI) confidence score was extracted from STRINGdb (http://string-db.org/). Clustering was performed using the R *hclust* function and the complete method. PPI clusters were defined visually and were retained when (i) they contained more than 10 genes and (ii) the mean score of the cluster was >0.4. Individual clusters were then explored for specific functions and components using STRINGdb.

The enrichment or depletion in histone modification was assessed using data from Pokholok et al., 2005 (http://younglab.wi.mit.edu/nucleosome/DataDownload.html). In brief, 8 histone modifications (3 acetylations: H3K9ac, H3K14ac, and H4ac and 5 methylations: H3K4me1, H3K4me2, H3K4me3, H3K36me3, and H3K7me3) were searched for in up- or downregulated genes in the mutant, as well as in non-significant genes between mutant and wildtype. The average ratios of acetylation and methylation in up- and downregulated genes were compared to those of non-significant genes (Kruskal–Wallis tests, *p*-value < 0.05). *p*-values were corrected for multiple testing with the Benjamin–Hochberg adjustment. The fold enrichment in up- or downregulated genes compared to non-significant ones was also calculated.

### 4.7. Chemical Analyses

#### 4.7.1. Apolar Ester Analysis

In order to prevent the evolution of the ester content, all the wines were then frozen at −20 °C until analysis. Preliminary tests confirmed that freezing did not significantly affect the concentration of the investigated compounds (data not shown). Volatile compounds produced during fermentation in each condition were quantified. The concentrations of 14 esters (Table 1) (6 fatty acid ethyl esters, 4 higher alcohol acetates, and 4 alkylated ethyl esters) in each wine were determined using head space solid phase microextraction (HS-SPME) followed by gas chromatography–mass spectrometry (GC–MS), as previously described [49].

#### 4.7.2. Quantification of Volatile Acids and Hydroxylated Esters by Liquid–Liquid Extraction and GC/MS Analysis

Two hydroxylated esters, ethyl 3-hydroxybutanoate (3hC4C2) and ethyl 2-hydroxy-4-methylpentanoate or ethyl-leucate (2h4mC5C2), were assayed according to the method previously described [25]. The same method was also used to quantify 4 volatile linear acids (C3 to C8) as well as 3 alkylated acids (2mC3, 2mC4, and 3mC4). The monitored ions are listed in Appendix A. Compounds were characterized by comparing their linear retention indices and mass spectra with those of standards.

#### 4.7.3. Quantification of Hydroxylated Acids

Concentrations of 3-hydroxy butanoic acid (3hC4) and 2-hydroxy 4-methyl pentanoic acid (2h4mC5) were determined by gas chromatography–mass spectrometry (GC–MS) after derivatization steps, as previously described [25].

#### 4.7.4. Higher Alcohol Analyses

Fifty microliters of internal standard (4-methylpentan-2-ol 50 g/L in pure alcohol) were added to a 5 mL wine sample. The solution was homogenized in a vortex shaker, and a microvolume was injected in split mode into an HP-6890 gas chromatograph coupled with a flame ionization detector (FID) (injector temperature, 200 °C) using a CP-Wax 57 CB column (50 m × 0.32 mm i.d.; film thickness, 0.25 μm; Varian). The oven was programmed at 40 °C for the first minute and raised to 200 °C at 8 °C/min, with the final isotherm lasting 20 min. The carrier gas was hydrogen 5.5 (Air Liquide, France).

### 4.8. Sensory Analyses

#### 4.8.1. General Conditions

Sensory analyses were performed as described by Martin and de Revel [50]. The samples were evaluated at controlled room temperature (20 °C) in individual booths using covered, black glasses (NF V09-110, 1971) (AFNOR. Sensory analysis—apparatus—wine-tasting glass—ISO 3591. Anal. Sensorielle, 1977), containing about 50 mL liquid, coded with three-digit random numbers. Sessions lasted approximately 5 min.

#### 4.8.2. Sensory Panels

Panel 1 consisted of 22 judges, 8 males and 14 females, aged 26.4 ± 4.8 (mean ± SD). Panel 2 consisted of 17 judges, 6 males and 11 females, aged 27.6 ± 5.3 (mean ± SD). All panelists were research laboratory staff at Bordeaux University, selected for their experience in assessing fruity aromas in red wines.

#### 4.8.3. Discriminative Tests

Panel 1 was used for the triangular tests of the various aromatic reconstitutions (AR) (Appendix A). For these AR, wines fermented with Fx10-AE were supplemented in FAEE, HAA, AEE, and HEE up to concentrations found in wines made with commercial yeast Fx10, and wines fermented with Fx10 were supplemented in C3C2 and AEE up to concentrations found in wines made with Fx10-AE, in order to have the same ester concentrations in both supplemented wines. These supplemented wines were then compared to one another (Appendix A).

For each triangular test, three numbered samples were presented in random order: two identical and one different. The wines were presented to the panel in duplicate during the same session. Each judge used direct olfaction to identify the sample perceived as different in each test and gave an answer, even if he or she was not sure. The results of all the triangular tests were statistically analyzed, based on the binomial law corresponding to the distribution of answers in this type of test.

#### 4.8.4. Descriptive Testing Methods

First, sensory profiles were realized by Panel 2 to precisely describe the nature of the differences observed during the first triangular tests. In a second phase, other sensory profiles for the aromatic reconstitutions (Appendix A) were evaluated by Panel 1. The wines were presented to the panel in duplicate during the same session to evaluate intensities for overall aroma and red-, black-, fresh-, and jammy-fruit characteristics. These aromatic descriptors were selected as the most typical of red wines from the Bordeaux area [51]. For each sample, the subject rated the intensity of these descriptors on a continuous 10 cm scale printed on paper, labeled “no odor perceived” on the left and “very intense” on the right.

### 4.9. Statistical Analysis

Quantitative data were analyzed using the R software with the following tests: A one or two-way analysis of variance (ANOVA) was applied to a linear model using the *car* package. The conditions of application (i.e., homogeneity of variance and normality of residuals) were controlled using Levene’s test and the Shapiro–Wilk test, respectively. Kruskal–Wallis tests were applied using the *agricolae* package, allowing the post hoc test based on Fisher’s LSD criterium with Benjamini Hochberg (BH) multiple test correction.

## 5. Conclusions

This functional genetic study serves to reassess the biosynthesis of different classes of esters in the context of alcoholic fermentation. As well as the four main esterases previously described, we demonstrated the role of two MAGLases (Yju3p and Mgl2p) in the biosynthesis of substituted ethyl esters. The multiple depletion of such esterases has unsuspected consequences on cell viability and on transcriptome regulation, as it impacts nitrogen and lipid metabolism as well as chromatin modification. The physiological role of ester biosynthesis has been debated previously. Several authors proposed detoxifying cytoplasm from medium-chain fatty acid and/or higher alcohols [12,52], but this hypothesis has never been confirmed experimentally. Ester production also plays an ecological role as it promotes yeast dissemination and migration by attracting insects [53]. In this study, we argue that the biosynthesis of esters could also be involved in acetyl-CoA homeostasis in fermentative conditions.

## Figures and Tables

**Figure 1 ijms-22-04026-f001:**
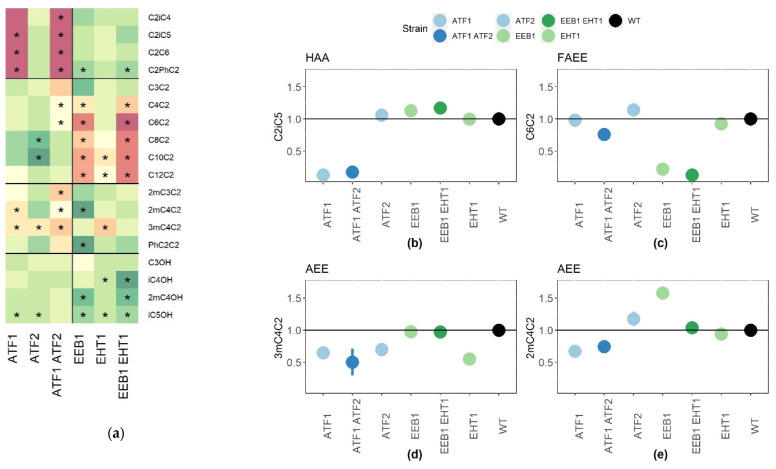
Depletion effect of the activities of acetyl-transferase (Atf1p/Atf2p) and acyl-CoA:ethanol-O-acyl transferase (Eeb1p/Eht1p) on linear esters. (**a**) Modalities significantly different from the *wt* are shown for all the volatile compounds measured. The symbol * indicates which modality is significantly different from the *wt* according to a Kruskal–Wallis test followed by a post-hoc Fisher’s LSD analysis (α = 0.01). (**b**–**e**) Gene deletion effects for representative compounds of each ester class family, C2iC5 (isoamyl acetate), C6C2 (ethyl hexanoate), 3mC4C2 (ethyl 3methyl butanoate), and 2mC4C2 (ethyl 2methyl butanoate), respectively. The raw values of each deletion strain were normalized by the average value of the control strain Fx10. Each point and bar represents the mean and the standard error computed from 3 to 6 independent biological repetitions.

**Figure 2 ijms-22-04026-f002:**
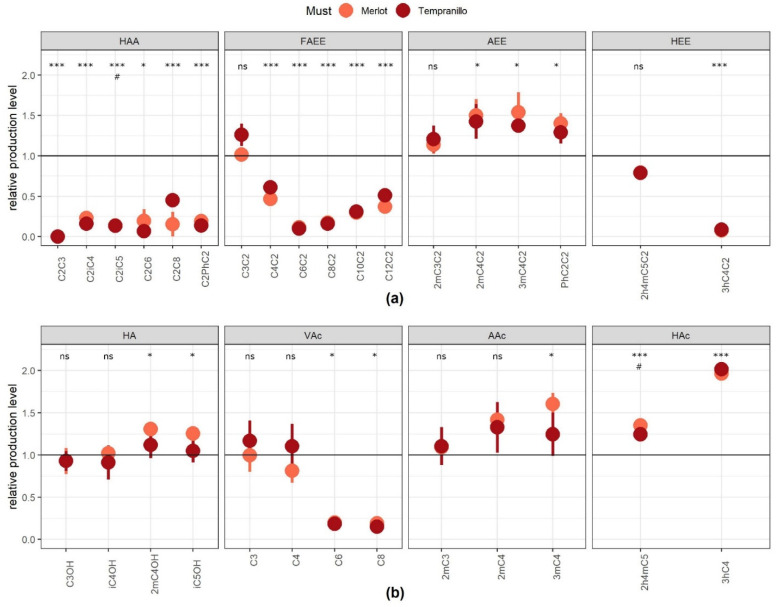
Relative production of esters and their metabolic precursors in a nearly “esterase free” strain. Raw values of the strain Fx10-ΔAE were normalized by the average value of the control strain Fx10. (**a**) The panel shows the relative production of each compound of the four ester families in both grape musts (Merlot and Tempranillo). (**b**) The panel shows the relative production of each metabolic precursor compound of each family in two grape musts. Each point and bar represents the mean and the standard error computed from 3 independent biological repetitions for each grape must. The symbols * and *** indicate which compounds have a significantly different production level with respect to the *wt* (post hoc HSD test based on the ANOVA, α = 0.05 and 0.001, respectively). The symbol # indicates which compounds have a significantly different production level according to the grape must origin (post hoc HSD test based on the ANOVA α = 0.05).

**Figure 3 ijms-22-04026-f003:**
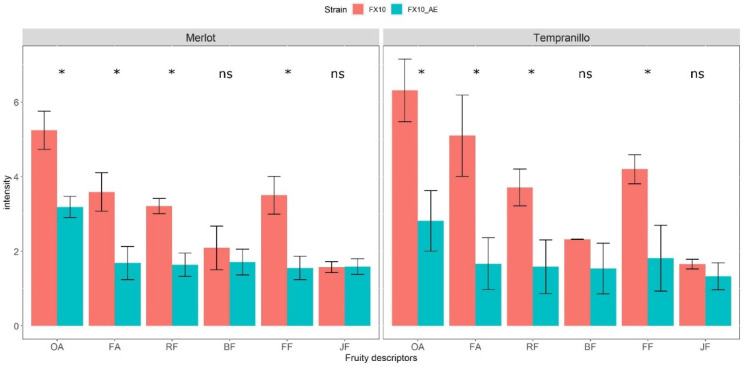
Sensory analysis of fruity perception of nearly “ester free” wines. Wines fermented by the yeast strains Fx10 and Fx10-ΔAE were evaluated by sensorial analysis by 26 panelists who evaluated the intensity of different fruity descriptors: OA: overall aroma, FA: fermentative aroma, RF: red fruit, BF: black fruit, FF: fresh fruit, JF: jammy fruit. The average intensity value for each descriptor and each strain is represented for the Merlot and Tempranillo wines. A Wilcoxon test was carried out to identify the difference of intensity between the two strains; the symbol * indicates a significant difference (α = 0.05).

**Figure 4 ijms-22-04026-f004:**
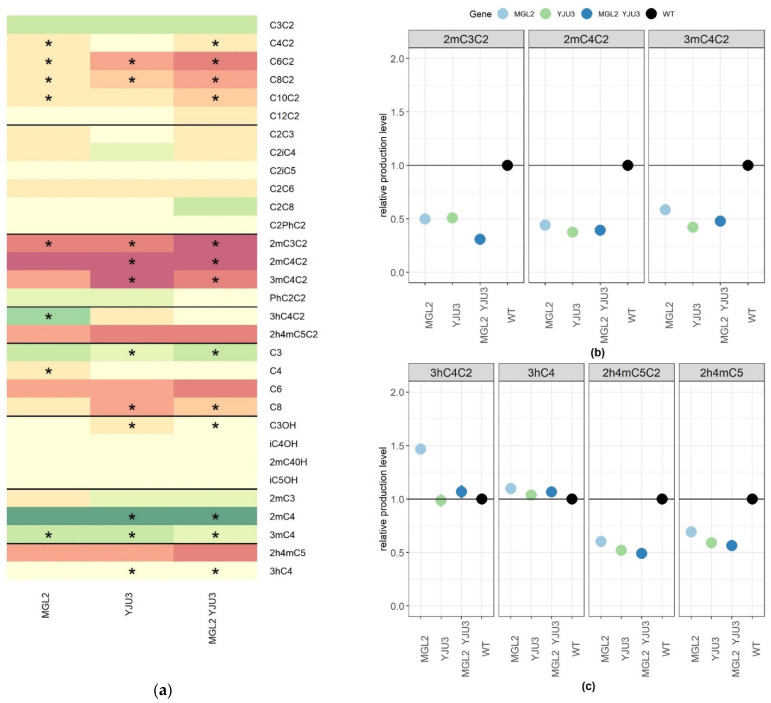
Effect of mono-acyl glycerol lipases (Mgl2p/Yju3p) inactivation on substituted esters and their relative metabolic precursors. The raw values of each deletion strain (*MGL2* and/or *YJU3*) were normalized by the average value of the control strain Fx10. (**a**) Modalities significantly different from the *wt* are shown for all the volatile compounds measured. (**b**) Deletion effect for significantly impacted AEE. (**c**) Deletion effect for HEE and their relative metabolic precursors. Each point and bar represents the mean and the standard error computed from 4 independent biological repetitions measured in two grape juices. The symbols * indicate which modality is significantly different from the *wt* according to a post hoc HSD test (α = 0.001).

**Figure 5 ijms-22-04026-f005:**
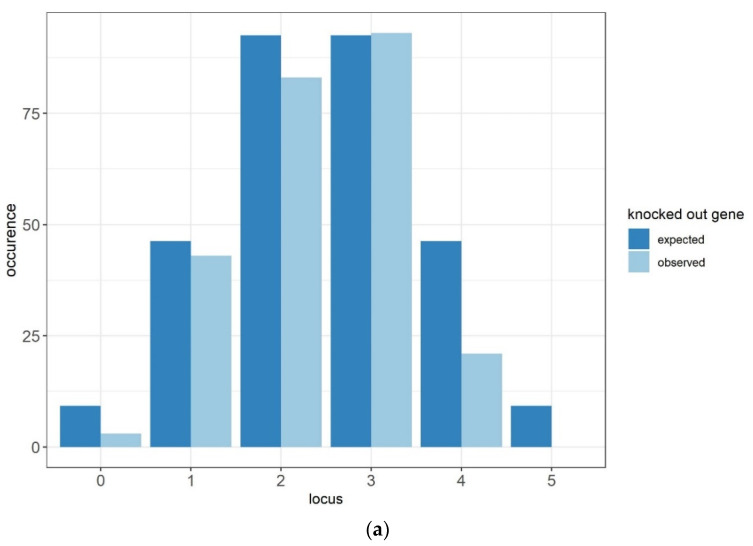
Genetic interactions between esterase genes impair the development of a multiple deletion strain. (**a**) Observed (light blue) and expected (dark blue) occurrences of meiotic progeny of the Fx10-ΔAEM (*ATF1/*Δ*atf1*, *ATF2/*Δ*atf2*, *EEB1/*Δ*eeb1*, *EHT1/*Δ*eht1*, *MGL2/*Δ*mgl2*) harboring 0 to 5 knocked-out genes. (**b**) Frequencies of meiotic segregants of the hybrids H1xH5 and H1xH6 showing a double deletion for the genes *ATF1*, *ATF1*, *EEB1*, *EHT1*, and *MGL2*.

**Figure 6 ijms-22-04026-f006:**
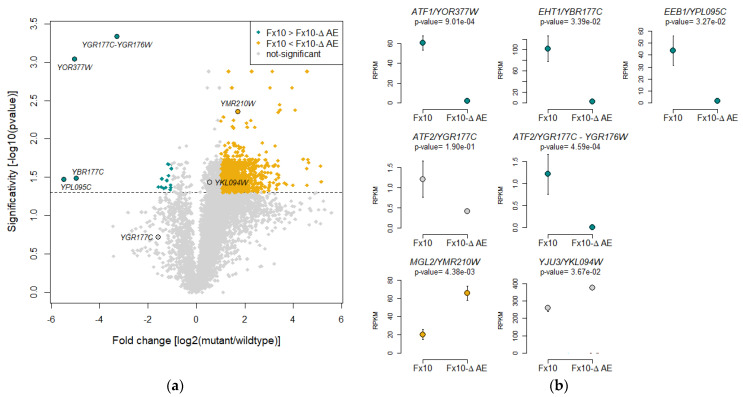
RNA-seq analysis of the Fx10-ΔAE mutant. (**a**) Volcano plot showing differentially expressed genes in the Fx10-ΔAE mutant compared to the Fx10 wildtype strain. A total of 6287 genes were considered, of which 1124 were DEGs. Yellow and blue-green dots show the 22 and 1102 genes downregulated or upregulated in the mutant, respectively. Gray dots represent the 5163 non-significant genes. (**b**) Expression patterns for genes of interest in the Fx10 wildtype and the Fx10-ΔAE mutant. RPKM stands for reads per kilobase million. Error bars represent standard errors. *p*-Values were corrected for False Discovery Rate (FDR) multiple tests.

**Figure 7 ijms-22-04026-f007:**
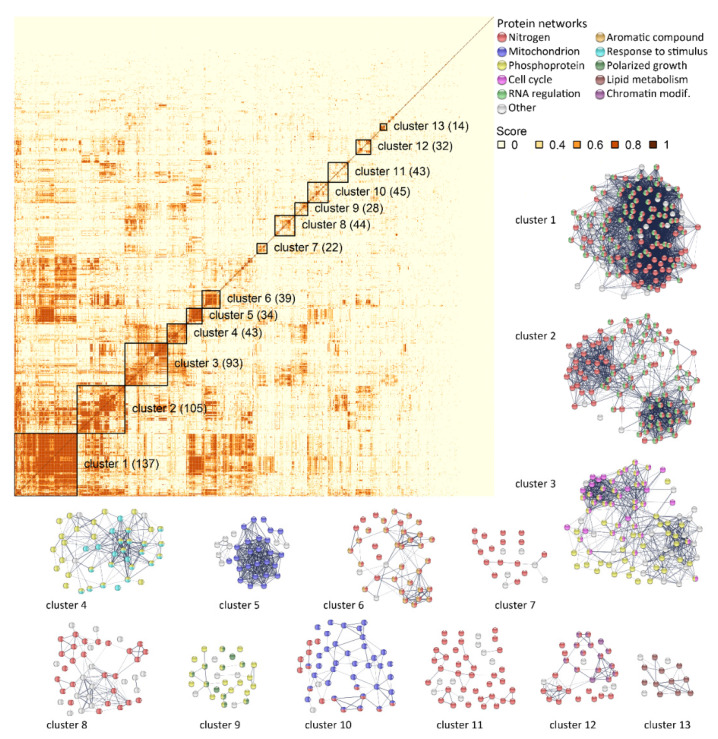
STRING protein–protein interaction analysis. Clustered pairwise correlation matrix (top left) of the 1124 differentially regulated genes in the Fx10-ΔAE mutant. For each pair of genes, a protein–protein interaction (PPI) confidence score was extracted from STRINGdb (http://string-db.org/). A total of 13 clusters are visually defined (the number of proteins per cluster is shown in brackets), and the STRING PPI network connectivity is represented. For each cluster, the main functional enrichment(s) of the networks (biological process from gene ontology, local network cluster from STRING, or annotated keywords from UniProt) are represented by different colors.

**Figure 8 ijms-22-04026-f008:**
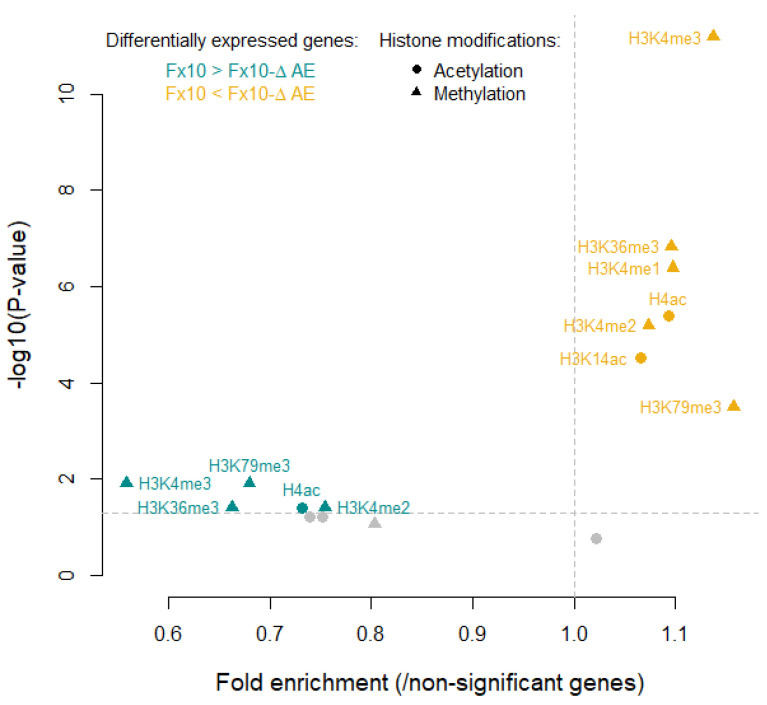
Enriched or depleted histone modifications in the Fx10-ΔAE mutant. Possible histone modifications (3 acetylations and 5 methylations) were sought for all genes. The average ratios of acetylations and methylations in up- and downregulated genes were compared to those of non-significant genes (Kruskal–Wallis tests). *p*-values were corrected for multiple testing with Benjamin–Hochberg adjustment. Not-significantly enriched/depleted histone modifications are shown in gray, whereas significant ones (*p*-value < 0.05) are colored. The histone modifications data are available from [28].

**Figure 9 ijms-22-04026-f009:**
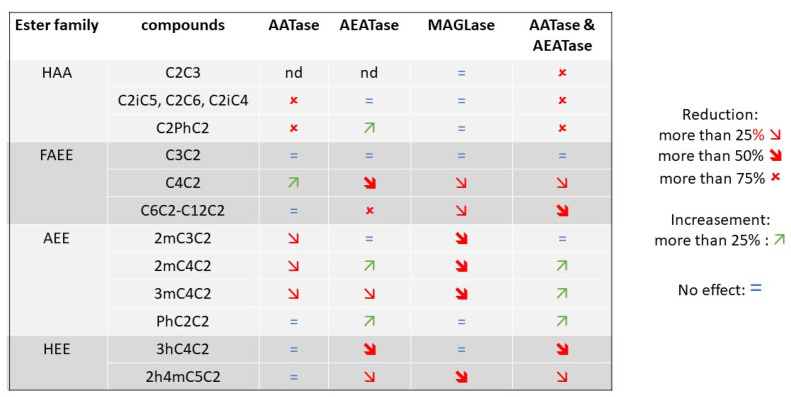
Relative contributions of AATses, AEATases, and MAGLases to the biosynthesis of volatile esters in a winemaking context.

**Table 1 ijms-22-04026-t001:** Chemical compounds assayed by family.

Esters			Metabolic Precursors		
Compounds	Family	Abbreviation	Compounds	Family	Abbreviation
ethyl propanoate	Fatty acid ethyl esters (FAEE)	C3C2	propanoic acid	Volatile acids (VAc)	C3
ethyl butanoate		C4C2	butanoid acid		C4
ethyl hexanoate		C6C2	hexanoic acid		C6
ethyl octanoate		C8C2	octanoic acid		C8
ethyl decanoate		C10C2			
ethyl dodecanoate		C12C2			
propyl acetate	Higher alcohol acetates (HAA)	C2C3	propan-1-ol	Higher alcohols (HA)	C3OH
2-methylpropyl acetate		C2iC4	2-methylpropan-1-ol		iC4OH
3-methylbutyl acetate (isoamyl acetate)		C2iC5	3-methylbutanol (isoamyl alcohol)		iC5OH
hexyl acetate		C2C6			
octyl acetate		C2C8			
2-phenylethyl acetate		C2PhC2			
ethyl 2-methylpropanoate	Alkylated ethyl esters (AEE)	2mC3C2	2-methylpropanoic acid	Alkylated acids (AAc)	2mC3
ethyl 2-methylbutanoate		2mC4C2	2-methylbutanoic acid		2mC4
ethyl 3-methylbutanoate		3mC4C2	3-methylbutanoic acid		3mC4
ethyl phenylacetate		PhC2C2			
ethyl 2-hydroxy-4-methyl-pentanoate	Hydroxylated ethyl esters (HEE)	2h4mC5C2	ethyl 2-hydroxy-4-methyl-pentanoic acid	Hydroxylated acids (HAc)	2h4mC5
ethyl 3-hydroxy-butanoate		3hC4C2	ethyl 3-hydroxy-butanoic acid		3hC4

**Table 2 ijms-22-04026-t002:** Yeast strains used.

Strain	Background	Genotype	Description	Origin
Y31674	BY4743	*BY4743;Mata/α;his3*Δ*1/his3*Δ*1;leu2*Δ*0/leu2*Δ*0;lys2*Δ*0/LYS2;MET15/met15*Δ*0,ura3*Δ*0/ura3*Δ*0;YOR377::kanMx4/YOR377::kanMx4*	*ATF1* deletion	Euroscarf
Y34807	BY4743	*BY4743;Mata/α;his3*Δ*1/his3*Δ*1;leu2*Δ*0/leu2*Δ*0;lys2*Δ*0/LYS2;MET15/met15*Δ*0,ura3*Δ*0/ura3*Δ*0;YGR177::kanMx4/YGR177::kanMx4*	*ATF2* deletion	Euroscarf
Y33317	BY4743	*BY4743;Mata/α;his3*Δ*1/his3*Δ*1;leu2*Δ*0/leu2*Δ*0;lys2*Δ*0/LYS2;MET15/met15*Δ*0,ura3*Δ*0/ura3*Δ*0;YRR177::kanMx4/YRR177::kanMx4*	*EEB1* deletion	Euroscarf
Y32157	BY4743	*BY4743;Mata/α;his3*Δ*1/his3*Δ*1;leu2*Δ*0/leu2*Δ*0;lys2*Δ*0/LYS2;MET15/met15*Δ*0,ura3*Δ*0/ura3*Δ*0;YPL095::kanMx4/YPL095::kanMx4*	*EHT1* deletion	Euroscarf
Y30796	BY4743	*BY4743;Mata/α;his3*Δ*1/his3*Δ*1;leu2*Δ*0/leu2*Δ*0;lys2*Δ*0/LYS2;MET15/met15*Δ*0,ura3*Δ*0/ura3*Δ*0;YMR210w::kanMx4/YMR210w::kanMx4*	*MGL2* deletion	Euroscarf
Y34943	BY4743	*BY4743;Mata/α;his3*Δ*1/his3*Δ*1;leu2*Δ*0/leu2*Δ*0;lys2*Δ*0/LYS2;MET15/met15*Δ*0,ura3*Δ*0/ura3*Δ*0;YKL094w::kanMx4/YKL094w::kanMx4*	*YJU3* deletion	Euroscarf
Fx10 HO/ho::HYG	Fx10	*Fx10; Mata/* *α* *, HO/ho::HYG*	Diploid homozygous	ISVV collection
Fx10-ΔA1	Fx10	*Fx10; Mata/α; HO/ho::HYG; YOR377::kanMx4/YOR377::kanMx4*	*ATF1* deletion	this study
Fx10-ΔA2	Fx10	*Fx10; Mata/α, HO/ho::HYG; YGR177::kanMx4/YGR177::kanMx4*	*ATF2* deletion	this study
Fx10-ΔE1	Fx10	*Fx10; Mata/α, HO/ho::HYG; YRR177::kanMx4/YRR177::kanMx4*	*EEB1* deletion	this study
Fx10-ΔE2	Fx10	*Fx10; Mata/α, HO/ho::HYG; YPL095::kanMx4/YPL095::kanMx4*	*EHT1* deletion	this study
Fx10-ΔA12	Fx10	*Fx10; Mata/α, HO/ho::HYG; YOR377::kanMx4/YOR377::kanMx4; YGR177::kanMx4/YGR177::kanMx4*	*ATF1*, *ATF2* deletion	this study
Fx10-ΔE12	Fx10	*Fx10; Mata/α, HO/ho::HYG; YRR177::kanMx4/YRR177::kanMx4; YPL095::kanMx4/YPL095::kanMx4*	*EEB1, EHT1* deletion	this study
Fx10-ΔAE	Fx10	*Fx10; Mata/α, HO/ho::HYG; YOR377::kanMx4/YOR377::kanMx4; YGR177::kanMx4/YGR177::kanMx4; YRR177::kanMx4/YRR177::kanMx4; YPL095::kanMx4/YPL095::kanMx4*	*ATF1, ATF2, EEB1, EHT1* deletion	this study
Fx10-ΔM	Fx10	*Fx10; Mata/α, HO/ho::HYG, YMR210w::kanMx4/YMR210w::kanMx4*	*MGL2* deletion	this study
Fx10-ΔY	Fx10	*Fx10; Mata/α, HO/ho::HYG; ;YKL094w::kanMx4/YKL094w::kanMx4*	*YJU3* deletion	this study
Fx10-ΔME	Fx10	*Fx10; Mata/α, HO/ho::HYG, YMR210w::kanMx4/YMR210w::kanMx4, YRR177::kanMx4/YRR177::kanMx4; YPL095::kanMx4/YPL095::kanMx4*	*MGL2, EEB1, EHT1* deletion	this study
Fx10-ΔMY	Fx10	*Fx10; Mata/α, HO/ho::HYG, YMR210w::kanMx4/YMR210w::kanMx4, ;YKL094w::kanMx4/YKL094w::kanMx4*	*MGL2, YJU3* deletion	this study
Fx10-ΔAEM	Fx10	*Fx10; Mata/a, HO/ho::HYG; YOR377 /YOR377::kanMx4; YGR177 /YGR177::kanMx4; YRR177 /YRR177::kanMx4; YPL095 /YPL095::kanMx4; YMR210w /YMR210w::kanMx4*	Heterozygous hybrid for *ATF1, ATF2, EEB1, EHT1, MGL2* deletion	this study
H1xH5	Fx10	*Fx10; Mata/a, HO/ho::HYG; YOR377 /YOR377::kanMx4; YGR177 ::kanMx4/YGR177::kanMx4; YRR177::kanMx4/YRR177::kanMx4; YPL095 /YPL095::kanMx4; YMR210w /YMR210w::kanMx4*		this study
H1xH6	Fx10	*Fx10; Mata/a, HO/ho::HYG; YOR377 /YOR377::kanMx4; YGR177/YGR177::kanMx4; YRR177::kanMx4/YRR177::kanMx4; YPL095::kanMx4/YPL095::kanMx4; YMR210w /YMR210w::kanMx4*		this study
Y31674	BY4743	*BY4743;Mata/α;his3*Δ*1/his3*Δ*1;leu2*Δ*0/leu2*Δ*0;lys2*Δ*0/LYS2;MET15/met15*Δ*0,ura3*Δ*0/ura3*Δ*0;YOR377::kanMx4/YOR377::kanMx4*	*ATF1* deletion	Euroscarf
Y34807	BY4743	*BY4743;Mata/α;his3*Δ*1/his3*Δ*1;leu2*Δ*0/leu2*Δ*0;lys2*Δ*0/LYS2;MET15/met15*Δ*0,ura3*Δ*0/ura3*Δ*0;YGR177::kanMx4/YGR177::kanMx4*	*ATF2* deletion	Euroscarf

**Table 3 ijms-22-04026-t003:** Summary of haplotypes tested.

Pedigree	Ploidy	Haplotype ^1^	Viable	Unviable	Number of Functional Copies				
					ATF1	ATF2	EEB1	EHT1	MGL2
Fx10-Delta5-progeny	n	H1	5	0	0	0	0	0	2
	n	H2	1	0	0	2	2	2	0
	n	H3	1	0	0	2	2	0	0
	n	H4	1	0	2	0	0	0	0
	n	H5	2	0	2	2	0	0	0
	n	H6	2	0	2	0	0	2	0
F2-hybrids	2n	H1/H2	0	20	0	1	1	1	1
	2n	H1/H3	0	20	0	1	1	0	1

^1^ For viable strains, haplotypes were verified by PCR; for the unviable background they were inferred from parental genotype.

## Data Availability

The data that support the findings of this study are available from the corresponding author upon reasonable request.

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
