# Peer review of "Metabolic, Organoleptic and Transcriptomic Impact of Saccharomyces cerevisiae Genes Involved in the Biosynthesis of Linear and Substituted Esters"

_ijms, 2021, doi:10.3390/ijms22084026_

Round 1

Reviewer 1 Report

The authors showed that although AFT1, AFT2, EEB1 and EHT1 had a moderate effect on the biosynthesis of Substituted Ethyl Esters, two MAGLase (Yju3p and Mgl2p) served a more important role on their biosynthesis.

The strategy for the research and analytical method used in the present study were reasonable. Existing literature is properly cited.

However, the manuscript is not clear and well organized. I would like to recommend that you focus on what you want to convey and recalibrate. And several English grammar mistakes were observed. The proofreading of the English sentences should be required.

It might be valuable information for the field of research of wine production, and the scientific impact and novelty of this article are potentially suitable for the publication after major-revision.

Specific suggestions:

Line 25

“help” is better instead of “complete”

Line 26-28 “In order to…..routes”

I don't understand the importance of this information. I thought it should be deleted.

Grammar mistakes:

Line 24 “depends on” is better

Line 197 “They concentration” may be “their concentration”.

Line 214 “both hydroxylated but also alkylated”

Line 450 “that were never considered before”

Line 520 “By fermenting two macerated grape juices with the strain FX10” is better.

Lin 543-546 Please insert “that” between “idea” and “Mgl2p”.

Author Response

Reviewer 1

Open Review

English language and style

(x) Extensive editing of English language and style required
( ) Moderate English changes required
( ) English language and style are fine/minor spell check required
( ) I don't feel qualified to judge about the English language and style

Yes

Can be improved

Must be improved

Not applicable

Does the introduction provide sufficient background and include all relevant references?

( )

(x)

( )

( )

Is the research design appropriate?

( )

(x)

( )

( )

Are the methods adequately described?

( )

(x)

( )

( )

Are the results clearly presented?

( )

( )

(x)

( )

Are the conclusions supported by the results?

( )

( )

(x)

( )

Comments and Suggestions for Authors

The authors showed that although AFT1, AFT2, EEB1 and EHT1 had a moderate effect on the biosynthesis of Substituted Ethyl Esters, two MAGLase (Yju3p and Mgl2p) served a more important role on their biosynthesis. The strategy for the research and analytical method used in the present study were reasonable. Existing literature is properly cited.

However, the manuscript is not clear and well organized. I would like to recommend that you focus on what you want to convey and recalibrate..

We clarified the two scopes of this study. First, we aim to find out the enzymatic activities controlling the biosynthesis of substituted esters. This goal was reached by following progressive functional analysis: by evaluating the deletion effect of AATase and AEATses, then by exploring the role of MGL2 and YJU3 genes.

The second aim was the evaluation of sensory impact of such genes by getting an “esterase free” yeast strain. Since this goal was not perfectly achieved, we provided experimental facts suggesting the existence of genetic incompatibilities that were explored by carrying out a transcriptomic analysis. According to the reviewer comment we reworded the last part of the introduction lines 79-90

And several English grammar mistakes were observed. The proofreading of the English sentences should be required

The final manuscript was reviewed by a professional and independent proofreader from an international company and many errors were corrected as illustrated by the marked version provided

It might be valuable information for the field of research of wine production, and the scientific impact and novelty of this article are potentially suitable for the publication after major-revision.

Specific suggestions:

Line 25

“help” is better instead of “complete”

 Corrected accordingly

Line 26-28 “In order to…..routes”

I don't understand the importance of this information. I thought it should be deleted.

We agree, the sentence was not very informative as is. We reworded the summary taking into account the general comments of the reviewer.

Grammar mistakes:

Line 24 “depends on” is better

Corrected accordingly.

Line 197 “They concentration” may be “their concentration”.

We corrected this typo

Line 214 “both hydroxylated but also alkylated”

Rephrased accordingly.

Line 450 “that were never considered before”

Rephrased accordingly.

Line 520 “By fermenting two macerated grape juices with the strain FX10” is better.

Corrected

Lin 543-546 Please insert “that” between “idea” and “Mgl2p”.

Done

Reviewer 2 Report

The proposed manuscript represents an interesting advance of state-of-the-art regarding biosynthesis of esters. I consider the experiments to be well performed, the results adequate and the conclusions relevant. 

Some minor comments:

  • Figures: the quality is so bad that for the majority of them I could not do a proper assessment
  • line 71-72 and Table 1 - authors should justify the list of genes and  chemical compounds assessed - why include these and not others that were of possible inclusion
  • - line 100 - "Quantitative" instead of "quantitative"
    - line 100-101 - please rephrase
  • line 102 - drastically should be substituted by other more scientific word
  • Section 4.3 - more experimental details about fermentation trials should be given.

As a final more general comment, it is my advice that the manuscrit could be improved by the inclusion of additional references for other similar works. The total number of citations is not sufficient enough, in my opinion, for a manuscript with this length and this amount of results/experiments. However, it is up to authors to improve this. It didn´t influence my revision.

  •  

Author Response

Open Review

English language and style

( ) Extensive editing of English language and style required
( ) Moderate English changes required
(x) English language and style are fine/minor spell check required
( ) I don't feel qualified to judge about the English language and style

Yes

Can be improved

Must be improved

Not applicable

Does the introduction provide sufficient background and include all relevant references?

(x)

( )

( )

( )

Is the research design appropriate?

(x)

( )

( )

( )

Are the methods adequately described?

( )

(x)

( )

( )

Are the results clearly presented?

(x)

( )

( )

( )

Are the conclusions supported by the results?

(x)

( )

( )

( )

Comments and Suggestions for Authors

The proposed manuscript represents an interesting advance of state-of-the-art regarding biosynthesis of esters. I consider the experiments to be well performed, the results adequate and the conclusions relevant. 

Some minor comments:

  • Figures: the quality is so bad that for the majority of them I could not do a proper assessment,

Please accept our apologize for this bad quality, we magnified the size of annotations especially for the figure 4. The quality of the figure are good but there is a problem in their insertion as part of a table in the world document. This will be fixed at the production step.

  • line 71-72 and Table 1 - authors should justify the list of genes and  chemical compounds assessed - why include these and not others that were of possible inclusion

Concerning genes : EAT1 was not considered as explained line 105

Concerning aromas this was shortly explained line 95:97, most of the esters found in wines are quantified at least those assayed by the laboratory.

  • - line 100 - "Quantitative" instead of "quantitative"
  • - line 100-101 - please rephrase

We rephrased this sentence.

  • line 102 - drastically should be substituted by other more scientific word

Changed by “strongly.”

  • Section 4.3 - more experimental details about fermentation trials should be given.
  •  

We add details on stirring, inoculation and CO2 monitoring in order to clarifier some imprecisions, the fermentation procedure refers to x methodological articles that deaply details how fermentations are carried out in the lab.

As a final more general comment, it is my advice that the manuscrit could be improved by the inclusion of additional references for other similar works.

The total number of citations is not sufficient enough, in my opinion, for a manuscript with this length and this amount of results/experiments. However, it is up to authors to improve this. It didn´t influence my revision.

It is true that many references and reviews are available on this subject:

Concerning the introduction we add some references for :

(i) higher alcohol biosynthesis another review and two original articles (line 49)

(ii) Multi genetic determinism of ester metabolism (3 new references at line 45)

(iii) Aromatic contribution or esters in beverages: an additional reference (line 39)

(iv) AATase activity (1 additional reference line 50-51

Round 2

Reviewer 1 Report

I have no further comments.